# Multi-Annual Dynamics of a Coastal Groundwater System with Soil-Aquifer Treatment and Its Impact on the Fate of Trace Organic Compounds

Quentin Guillemoto [1,2,*], Géraldine Picot-Colbeaux [1], Danièle Valdes [2], Nicolas Devau [1], Charlotte Thierion [3], Déborah Idier [1], Frédéric A. Mathurin [1], Marie Pettenati [1], Jean-Marie Mouchel [2] and Wolfram Kloppmann [1]

1   BRGM, French Geological Survey, 45000 Orléans, France
2   UMR 7619 Metis, CNRS, EPHE, Sorbonne Université, 75252 Paris, France
3   ANTEA, 45000 Orléans, France
*   Correspondence: quentin.guillemoto@orange.fr; Tel.: +33-(0)2-38-64-31-37

**Abstract:** The combination of managed aquifer recharge (MAR) with soil-aquifer treatment (SAT) has clear advantages for the future sustainable quality and quantity management of groundwater, especially when using treated wastewater. We built a MARTHE flow and transport model of an MAR–SAT system located in a near-shore sand aquifer, for quantifying the influence of environmental factors (climate, tides, and operational conditions) on the coastal hydrosystem with regard to the fate of trace organic compounds (TrOCs). The simulations show the impact of these factors on flow rates and dilution, and thus, on the potential reactivity of TrOCs. The dilution of secondary treated wastewater (STWW) is variable, depending on the operations (feeding from infiltration ponds) and on shore proximity (dilution by saltwater). We show that, close to the ponds and during infiltration, the attenuation of TrOC concentrations can be explained by reactivity. At the natural outlet of the aquifer, the simulated average residence times ranged from about 70 to 500 days, depending upon seasonal dynamics. It is important to study TrOCs at site scale in order to anticipate the effect of natural variations on the SAT and on the fate of TrOCs.

**Keywords:** soil-aquifer treatment; numerical hydrogeological modelling; flow and transport; residence time; trace organic compounds; dilution; coastal area



## 1. Introduction

Managed aquifer recharge (MAR) is based on active aquifer management methods providing a local response to water scarcity, water security, water-quality degradation, aquifer depletion, and endangered ecosystems [1]. In coastal environments, soil-aquifer treatment (SAT) is one possible MAR technique that not only recharges the aquifer, but also provides natural purification of the incoming water; it takes advantage of the geochemical, physical, and biological processes occurring when its infiltrates the soil, unsaturated, and saturated zones of the aquifer.

Wastewater treatment plants (WWTPs) provide water that can be used for SAT systems [2]. Domestic or municipal treated wastewater is a source of water released daily by human activity, and therefore less subject to seasonal variations and climate change. Many SAT sites have been established, primarily in the coastal zone. They favor the infiltration of treated wastewater into the aquifer [3–6], commonly via infiltration ponds with intermittent recharge to maintain infiltration rates, and with soil oxygenation to renew the biological and chemical treatment capacity performed by natural processes in the soil and subsoil [2]. Infiltrated water recharges the aquifer, thus providing an additional treatment step for secondary treated wastewater (STWW) in response to local water stress or environmental issues (e.g., freshwater quantity and saltwater intrusion), and preventing direct discharge to surface water or the sea.

Many trace organic compounds (TrOCs), such as pharmaceuticals, personal care products, and pesticides, are present in wastewater and are not fully removed during passage through WWTPs [7,8]. They are then discharged into receiving waters via WWTP effluents and can reach various environmental compartments, such as groundwater [9] and rivers [10]. Some TrOCs and/or their metabolites persist in their active forms and can be toxicologically hazardous to the environment as well as to human health [11]. SAT can provide an additional treatment of TrOCs in treated wastewater [7,12], but many uncertainties remain concerning their fate in SAT systems. The primary mitigation mechanisms for TrOCs in SAT systems are degradation and sorption [2,12,13]. Sorption involves physical interactions that bind or slow down compounds in soil matrix materials (minerals, (dissolved) organic matter, etc.), affecting TrOC mobility in soil and aquifer. The degradation of TrOCs involves microorganisms, such as bacteria and/or fungi, which assimilate them via enzymes to maintain biomass.

TrOC behaviour in an SAT context depends on hydrogeological, geochemical, and biological conditions [13]. Their reactivity depends on many factors, resulting in a very wide range of values for the commonly used degradation coefficient [14], or sorption coefficient [15,16], depending on hydrodynamic conditions, soil properties (redox conditions, temperature, biomass, organic matter inputs, etc.) and physico-chemical properties of the molecules (e.g., pKa, charge, and hydrophobicity [13,17]). Over an annual time-scale, large changes in reactivity conditions (e.g., redox conditions, organic matter, and temperature) and system flow (e.g., groundwater dilution and flow velocities) can modify system performance.

To estimate and optimise the performance of MAR systems, numerical modelling is a suitable tool [3]. Numerical simulations can quantify the dilution evolution of the treated wastewater infiltrated in the SAT and the daily evolution of flow velocities, and thus, of residence time of the water over several years and at the scale of an aquifer. Such information is important for interpreting the possible evolutions of TrOC concentrations at site scale, and of the purification capacity of an SAT at aquifer scale.

In this paper, we focus on the specific case of an experimental coastal SAT system at Agon-Coutainville, France [18], where secondary treated wastewater (STWW) is alternately discharged into a dune aquifer via different infiltration ponds. Measurements of groundwater TrOC concentrations have been carried out since 2016 at the site [18–20]. Previous investigations on this experimental site have quantified the reactivity of molecules under specific experimental conditions [20].

The particularity of the Agon-Coutainville SAT site is that it is part of a coastal system where strong potentiometric variations are observed, mainly caused by tides, natural recharge, and infiltrated STWW volumes. Due to the uncertainties regarding the reactivity of TrOCs in an SAT and the possible variations in hydrodynamic conditions, understanding the fate of TrOCs in an SAT context at an annual scale is very complex. Identification of the reactive processes and their variations over time in such systems first requires precise characterisation of the hydrodynamic variations that modify residence times and dilution of the STWW infiltrated into the hydrosystem.

The objectives of the present study were twofold: (1) quantify the variations in residence time and dilution of the infiltrated water (STWW) in the Agon-Coutainville SAT site as a function of natural hydrodynamic forcing (natural recharge and runoff, and tides) and anthropogenic forcing (controlled recharge via the SAT); and (2) anticipate their effects on TrOC concentrations in the aquifer at the spatial scale of an aquifer (kilometre scale) and at a multi-year temporal scale. For the Agon-Coutainville SAT site, a numerical flow and transport model was developed for simulating variations in the flow and mixing velocities of infiltrated water (STWW) in the SAT at the hydrosystem scale. The numerical modelling tool reproduces the multi-year effects of natural and anthropogenic forcing that strongly influence the fate of TrOCs, in particular the variations in residence time and dilution of the treated wastewater caused by waters transiting through the system via tides and natural recharge.

Our work has improved the understanding of the fate of TrOCs in a dynamic coastal SAT system. The quantified residence and dilution times obtained from modelling are necessary tools for predicting the fate of TrOCs at the scale of a site, while distinguishing between reactive and flow-related processes in the abatement of TrOCs.

## 2. Materials and Methods

### 2.1. Study Area: Agon-Coutainville, Normandy, France

The Agon-Coutainville commune is located in Normandy (France), on the western coastline of the English Channel, between cape La Hague and the bay of Mont Saint Michel (Figure 1). The SAT system has been operational for over 20 years as a complement to the wastewater treatment plant (WWTP), to prevent direct discharge of secondary-treated wastewater (STWW) into the sea [18,20]. The WWTP is designed for a 35,300 population equivalent and uses activated sludge biological treatment. The STWW is transferred by gravity to the SAT and infiltrates via three infiltration ponds with a total surface area of 29,000 m$^2$ into a sand-dune aquifer (Figure 1). Each pond infiltrates STWW alternatively for four months and is not used for the rest of the year.

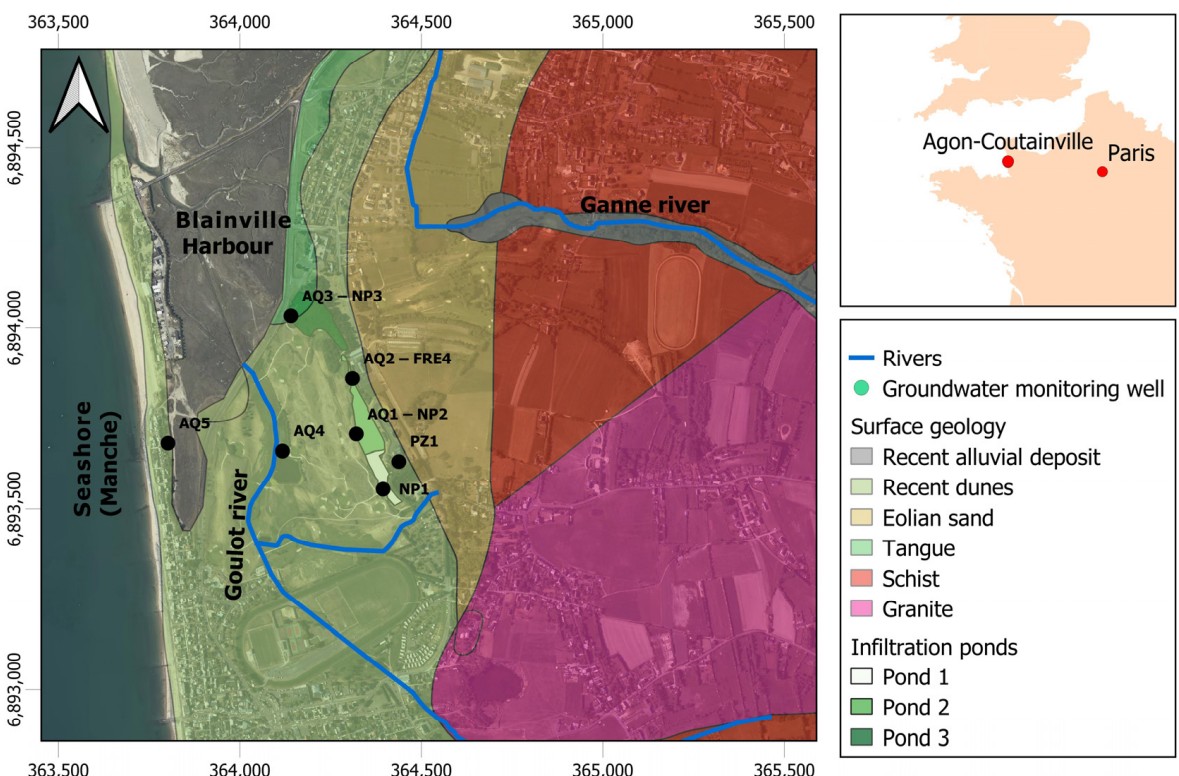

**Figure 1.** Location map of the SAT site at Agon-Coutainville (three infiltration ponds), observation wells (NP1, AQ1-NP2, AQ3-NP3, AQ2-FRE4, AQ4, AQ5, and Pz1), drainage network (blue lines), coastline and harbour, and surface geological formations.

The site is mainly underlain by Precambrian metamorphic rock (schist) surrounding Coutances granite (quartz diorite) (Figure 1), overlain by Quaternary aeolian sand and recent dunes [21]. These sands form the dune aquifer in which the SAT is developed. It lies on the metamorphic schist whose outcrops delimit it to the east. In the west, the sand-dune aquifer extends towards the sea.

Located on the coast at approximately 600 m from the sea (Figure 1), the SAT system is subject to tidal cycles. Here, the main cycles are: (1) semi-diurnal cycles with a period of 12.25 h; (2) bi-monthly cycles with a period of 14.8 days between periods of high and low tides; and (3) annual cycles linked to the solstices and equinoxes. Tidal variations are very significant, with a range of up to 13 m. The positions of the saltwater zone and the

freshwater/saltwater interface are poorly identified in the study area, in the absence of vertically discretised observations.

The climate is temperate and oceanic, with an average annual rainfall of 850 mm. Seasonal variations in sunshine and rainfall occur between winter (October to March being the wettest months) and summer (April to September being the driest months). Two streams cross the study area (Figure 1), the Ganne to the north and the Goulot to the south, both having their source in the metamorphic and granitic hills to the east and flowing into Blainville harbour.

### 2.2. Available and Acquired Data

The available data for the study site are summarised in Table 1. Surface geological information, described by Dupret et al., 1987, and spatialised information included in a geographic information system [22], were used. Vertical geological information is provided by the drilling of observation wells on the site (Figure 1). Spatial topographic information is provided by a 1 m and 25 m resolution digital elevation model (DEM), and differential GPS (Global Positioning System) data served to locate boreholes and infiltration ponds in three dimensions. The spatial position of streams was defined with the BDTOPO® (IGN, France) vector-based database. Daily precipitation and PET data for the period 2006 to 2021 are available from the Gouville-sur-Mer weather station (Météo-France), 10 km south of the study site.

**Table 1.** Available and acquired data for the Agon-Coutainville SAT site.

| Type of Data | Location | Measurements and Frequencies | Sources |
|---|---|---|---|
| Groundwater level | AQ1-NP2, AQ2-FRE4, AQ3-NP3, AQ4, AQ5, | Probes 15 min, 2017–2022 * | Projects AQUANES (April 2017 to April 2019) + EVIBAN (April 2019 to December 2022) |
| | NP1, AQ1-NP2, AQ2-FRE4, AQ3-NP3, AQ4, AQ5, PZ1 **, PTC6 ** | 9 Manual measurements from 2017 to 2021 | Projects AQUANES (6 from April 2017 to April 2019) + EVIBAN (3 from April 2019 to December 2022) |
| | NP1, PZ1 | Probes 15 min, 2020–2022 | EVIBAN project |
| GW analysis of $Cl^-$ | NP1, AQ1-NP2, AQ2-FRE4, AQ3-NP3, PZ1 | 12–14 per year | SAUR operator |
| | NP1, AQ1-NP2, AQ2-FRE4, AQ3-NP3, PZ1 | 9 field campaigns from 2016 to 2021 | Projects AQUANES (6 from April 2017 to April 2019) + EVIBAN (3 from April 2019 to December 2022) |
| STWW analysis of DBO5 | WWTP outlet | 24 per year | SAUR operator (2006 to 2022) |
| STWW Flow | WWTP outlet | Radar venturi channel, hourly | SAUR operator (2010–2022) |
| Meteorology | Gouville | Precipitation and PET, daily | METEO-FRANCE |
| Geology | Normandie | Maps—cross-section | [21,22] |
| | NP1, AQ1-NP2, AQ2-FRE4, AQ3-NP3, AQ4, AQ5, PZ1, (PTC6) | logs | SAUR Operator |
| Topography | Normandy Agon-Coutainville | Normalised to elevation (asl) 25 m Normalised to elevation (asl) 1 m | Region Region |
| Rivers | France | Map—streams—network | BDTOPO® (IGN, France) |

* Numerous missing data on continuous measurements. ** only 3 piezometric measurement campaigns for PTC6 and 2 for PZ1.

For the period 2010 to 2021, daily STWW flow data were acquired by radar monitoring of the levels in a Venturi channel, and biological oxygen demand (BOD5) was measured 24 times per year. Between 2016 and 2021, $Cl^-$ was analysed nine times in the STWW.

Eight observation wells screened across the thickness of the dune aquifer (Figure 1) provide access to groundwater measurements. Groundwater, STWW quality, and STWW flows have been monitored for over 20 years within the regulatory framework [18]. From 2016 to 2021, the site was the subject of nine measurement campaigns of groundwater quality and STWW as part of the AQUANES [19] and EVIBAN (Water JPI, 2022) research projects. The data used for our study were obtained from regulatory monitoring and these research projects. $Cl^-$ concentrations in groundwater were measured every month in wells NP1, AQ1-NP2, AQ2-FRE4, AQ3-NP3, and PZ1 between 2010 and 2021 from September to June, and every two weeks from July to August. From 2017 to 2021, water-level measurements [23] (based on pressure measurements corrected for atmospheric pressure) were acquired continuously in wells AQ1-NP2, AQ2-FRE4, AQ3-NP3, AQ4, and AQ5 [23].

In addition, continuous electrical conductivity measurements were performed in well AQ5. The initial measuring frequency was 15 min, but due to difficulties in obtaining complete data (such as remote transmission and equipment maintenance), gaps and frequency changes are present in the acquired time series. From 2017 to 2021, nine manual measurements of potentiometric levels were carried out in wells NP1, AQ1-NP2, AQ2-FRE4, and AQ3-NP3, and three more were performed at point PTC6. From 2020 to 2021, pressure and conductivity measurements were carried out in wells NP1 and PZ1 by CTD-Diver probes (Van Essen®, Delft, The Netherlands) at 15 min time steps (barometric probe located at NP1).

*2.3. Methodological Approach*

To quantify the flow rate, water balance and residence time of STWW through the SAT and into the dune aquifer, as well as its dilution by other waters transiting the aquifer (groundwater, natural recharge, and exchange with rivers and/or the sea), we used numerical modelling tools for simulating flow and solute transport within a hydrosystem under transient (space and time) conditions.

The main steps for quantifying the flow velocities and STWW proportions at the SAT study site at multi-year and dune/aquifer scales were: (1) the development of a conceptual flow and transport model of the hydrosystem; (2) its implementation into a numerical model with imposed boundary conditions for taking the major forcings into account; (3) calibration of the numerical model; (4) simulation of flow and groundwater mixing; (5) sensitivity analysis of the model; and (6) analysis of the simulation results considering the fate of TrOCs in the SAT.

The conceptual model describes the key hydrogeological processes in the study area, thus enabling simplifications and proposing boundary conditions [24]. Boundary conditions and geometry were set to allow for multi-year transient flow and solute transport calculations, considering the dynamics of external forcing and the operational dynamics of the SAT.

The hydrodynamic model was then calibrated from 2017 to 2021, using potentiometric-level time series observed at the site. Due to its nearly conservative nature and the chemically contrasting concentrations of waters (STWW, sea, natural recharge, and river), chloride ($Cl^-$) was chosen to calibrate the hydrodispersive parameters of the solute transport model. To distinguish the progression of STWW from other waters transiting through the aquifer, the calibrated model also used the concentration of a hypothetical non-reactive compound present at 100% only in the STWW infiltrated into the SAT. The calculated concentrations in the hydrosystem (initially at 0% for this compound), indicate the proportions of STWW, and thus, their dilution by other waters in the hydrosystem (saline intrusion, natural recharge, and stream water) in which this compound is absent.

The modelling identified flow directions, and calculated the hydrodynamic balance, water flow velocities in the pores, transit times along the main flow lines, and STWW proportions.

The average values and coefficients of variation (CV) of STWW velocities and proportions in each cell of the model represent the variations over a hydrogeological year (from 1 October to 30 September), according to:

$$CV = \frac{\sigma}{\mu} \tag{1}$$

where σ is the standard deviation and μ is the mean of the time series. The larger the CV coefficient, the greater the dispersion will be around the mean. The calculation was mapped for two hydrogeological years, 2017–2018 (low winter recharge) and 2020–2021 (high winter recharge). Temporal variations in flow velocity and dilution ratio were quantified from 2017 to 2021 along four main flow lines from the SAT–STWW infiltration ponds to the natural groundwater discharge points.

A sensitivity analysis then assessed the sensitivity of flow velocity and STWW proportion results to modelling choices (boundary conditions) or model calibration parameters. This identified the key parameters or model choices requiring more precise definition in order to reduce the uncertainty levels of the results.

## 3. Modelling

*3.1. Conceptual Model: The Local SAT System in Its Hydrodynamic Aquifer Context*

3.1.1. Coastal Sand Dune Aquifer

The topography of the study area reflects the geological nature of the subsoil. Ground elevation ranges from 50 metres above sea level (m.a.s.l.) for areas of outcropping metamorphic rock in the east, to 0 m.a.s.l. at sea level in the west, where sand forms the coastal dune aquifer (Figure 2). In the sand deposition zone (Figure 1), the topography of the aquifer top is sub-horizontal at 5–6 m.a.s.l., with local depressions where the SAT is located. This zone is separated from Blainville harbour by a dam road of 8–9 m.a.s.l. elevation. The topography defines several watersheds, including those of the Ganne and the Goulot, that cross the sandy deposits in their lower part and flow into the harbour (Figure 1). The surface of both watersheds in their lower part represents 2.2 km$^2$ of the study area, while their upstream metamorphic part covers 3.7 km$^2$ (Figure 2). The transfer of water from the upstream part to the sandy areas occurs via runoff and infiltration in the foothills or via streams.

The dune aquifer is covered by a very thin layer of vegetal soil. The aquifer bottom is formed by metamorphic bedrock, assumed to be impermeable. Although groundwater can circulate in fracture networks of bedrock schist, potentially causing low-flow springs [21], such processes were not observed on the study site. The dune aquifer is thickest in the west (9 m at piezometer AQ5, near the sea), decreases to 5 m close to the infiltration ponds [23,25], and disappears in the metamorphic foothills. The boreholes exhibit stratification, with coarse and shell sands overlying green clay resulting from erosion of the schist bedrock. "Tangue" deposits (fine clayey to peaty material with low permeability) may be present as discontinuous layers within the sands but was not observed in the various boreholes available at the site. The hydraulic conductivity (*K*) of coarse non-clayey sand is estimated between $10^{-2}$ and $10^{-5}$ m·s$^{-1}$, and between $10^{-5}$ and $10^{-9}$ m·s$^{-1}$ for fine clayey sand [26]. The effective porosity of coarse sand can range from 15 to 35% [27]. For aeolian sand with a smaller grain size, the effective porosity can decrease significantly [26].

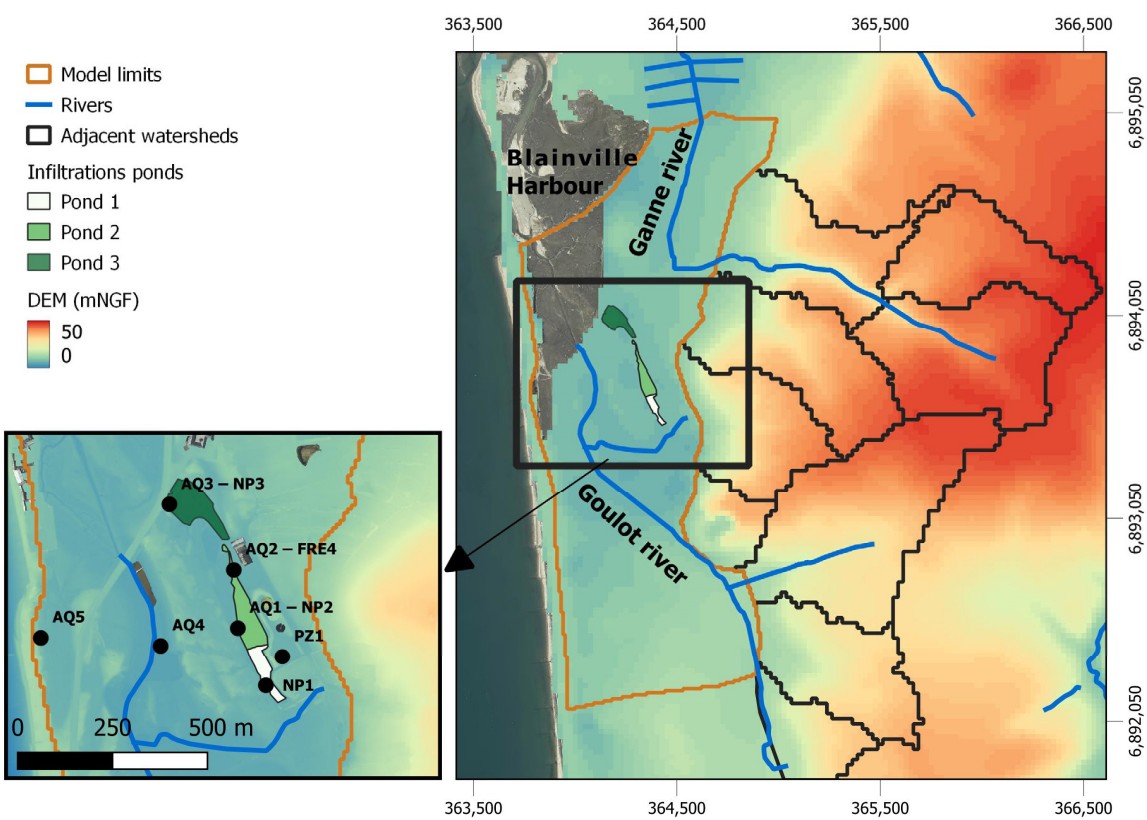

**Figure 2.** Topography (25 m DTM and 1 m DTM) and boundaries of the adjacent upstream watersheds and of the hydrogeological model, including the SAT system (infiltration ponds).

The dune aquifer contains an unconfined groundwater body; its potentiometric level decreases from east (5.1 ± 0.50 m.a.s.l. at PZ1; 45.1 ± 46 m.a.s.l. at NP1) to west (4.5 ± 0.17 m.a.s.l. at AQ4 and 3.1 ± 0.32 m.a.s.l. at AQ5). The resulting main flow gradient is about $10^{-3}$. On an annual scale, groundwater levels change by about 0.5 to 1.0 m due to variations in STWW infiltration rates in the SAT and to natural recharge of the aquifer. Vertical flow through the unsaturated zone is assumed to be negligible compared with horizontal flow, due to the thin unsaturated zone ranging from 0 to 1.5 m. The storage capacity in the unsaturated zone of the dune aquifer, for a porosity of 15% to 35%, is around 0.49 to 0.99 $Mm^3$. In some years, the water levels exceed the ground surface, especially in topographic depressions and the infiltration ponds (5.4 m.a.s.l. on average in the ponds). Groundwater then overflows and remains on surface before infiltrating back into the aquifer, causing localised flooding during winter. $Cl^-$ concentrations in the aquifer vary spatially and temporally through the mixing of natural recharge, runoff (freshwater), tidal water (brackish water), infiltrated STWW (variable salinity water), and river water (freshwater).

### 3.1.2. Natural Recharge and Runoff

Precipitation and ETP data (Supplementary Materials S1, Figure S1) show that natural recharge mainly occurs during winter (October to March), when the PET is low and precipitation is high, with discharge occurring from April to September when the groundwater table is at its lowest.

The quantity and distribution of natural recharge were estimated with a GARDENIA reservoir model [28] applied at the scale of the dune aquifer (surface 2.2 $km^2$, Figure 2) from daily rainfall and PET records. Due to the very low slopes of the dune aquifer and its sandy character, natural recharge is considered to be close to the effective rainfall with a negligible part of runoff. The model reservoir parameters (soil and unsaturated characteristics) were thus selected to minimise the amount of runoff and maximise natural

recharge (Supplementary Materials S1). Modelled "direct" natural recharge flow to the aquifer averages 0.84 mm/d (1707 $m^3$/d) over the period 2017 to 2021. Minimum flow was obtained for the 2018–2019 hydrogeological year, averaging 0.45 mm/d (1008 $m^3$/d), and maximum flow for 2019–2020 at 1.1 mm/d (2452 $m^3$/d). Cumulative annual recharge (280 mm) represents 34% of cumulative precipitation, which corresponds to the order of magnitude of natural recharge in mainland France [29].

In addition, runoff from adjacent watersheds (Figure 2) probably constitutes an additional natural indirect recharge of the dune aquifer, considering that part of the effective rainfall infiltrates in the foothills. Due to the presence of outcropping shale, considered impermeable, with significant slopes in the upstream parts of the watersheds (4% slopes), runoff reaches the dune aquifer either via the stream network or via the east edge of the aquifer. The $Cl^-$ concentration measured at PZ1 (Figure 1) is very low (36 mg/L on average); local infiltration of runoff from the upstream catchments on the east edge would prevent the progression of STWW infiltrated in the SAT towards PZ1, located between the east edge and the catchments. The amount and distribution of runoff was again estimated with the GARDENIA model. The model reservoir parameters were chosen this time to maximise runoff and limit recharge. The runoff rate, considered as additional natural recharge of the dune aquifer along its east edge, was estimated by considering the surface of the watersheds adjacent to the aquifer, which corresponds to the upstream metamorphic part (3.7 $km^2$). The calculated runoff flow (indirect recharge) averages 156 mm/y (1579 $m^3$/d) over 2017 to 2021, representing 84% of the direct recharge flows.

### 3.1.3. Streams and Rivers

The Ganne stream to the north is far from the SAT area of interest and crosses the dune aquifer over a very short distance. The Goulot stream, however, crosses the aquifer from south to north at right angles to the major east-to-west groundwater flow direction, probably impacting such flow from the SAT. The Goulot directly lies on the dune aquifer with a bed in which a deposit of organic matter and fine sediment can accumulate. The Goulot water level is approximately 4.7 m.a.s.l. and varies seasonally. The depth of the streambed varies between 30 cm and 60 cm, its width being between 4.0 and 5.5 m. The river is locally artificialized to pass under the southern part of town before being blocked by a dam, limiting its discharge into the harbour. Farther upstream, intermittent and smaller streams mainly flow from metamorphic rock areas into the Goulot, but no monitoring stations record stream flow. However, field observations indicate low flow of the Goulot (500–5000 $m^3$/day) and the streams are often dry, depending on weather conditions and groundwater levels.

### 3.1.4. Sea and Harbour Effects

Marine dynamics modify the groundwater levels and salinity along the coast. Water levels in observation wells AQ3-NP3, AQ4, and AQ5, near the sea and the harbour, are marked by tidal cycles, especially monthly "high water" and "low water" cycles, and by daily variations in low amplitude (about 0.1 m) in well AQ5 (Figure 3). The main variations in salinity linked to the sea were observed in wells AQ3-NP3 and AQ5. During spring tides, notably at equinoxes, the sea progressively fills Blainville harbour, bordering SAT pond 3 where the AQ3-NP3 well is located over a few metres, which can cause significant increases in $Cl^-$ (Figure 3).

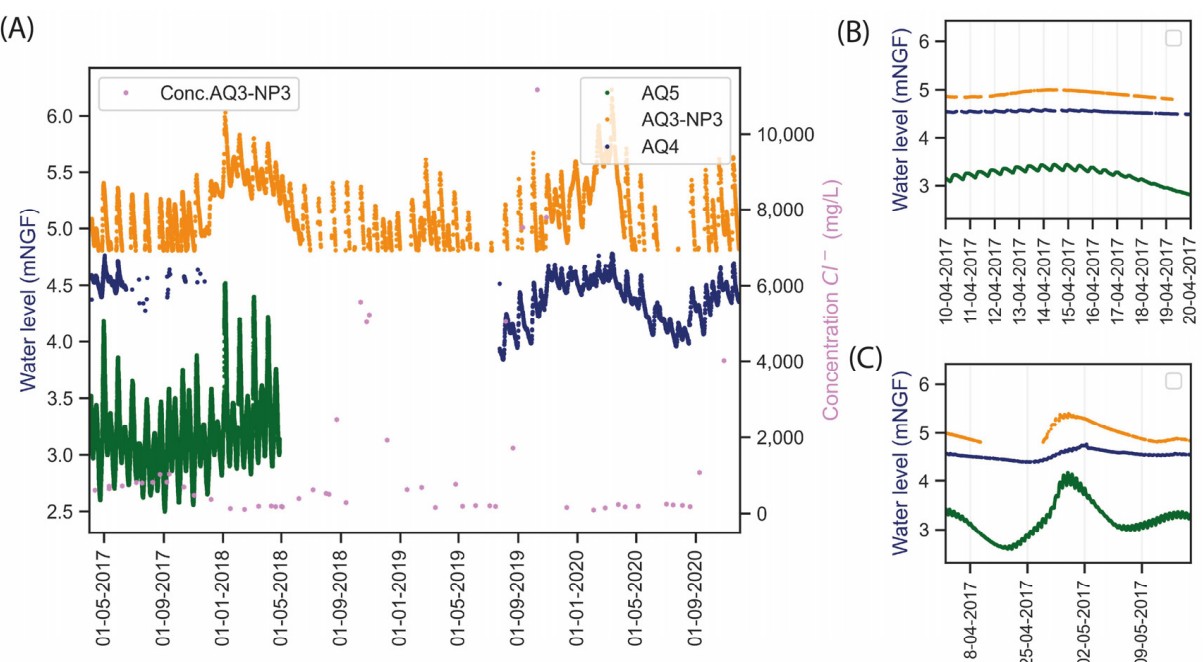

**Figure 3.** Potentiometric data from observation wells AQ5, AQ4, and AQ3-NP3, and Cl⁻ concentrations in AQ3-NP3 over the period from (**A**) 1 January 2017 to 5 January 2018, (**B**) 10 April 17 to 20 April 2017, (**C**) from 15 April 2017 to 15 May 2017.

Cl⁻ maxima of 11,180 mg/L were observed in AQ3-NP3, while electrical conductivity was highest in AQ5 with 12,481 ± 3678 μS/cm. Given the Cl⁻ concentration of around 19,000 mg/L in the sea [30], and the average Cl⁻ concentrations of 36 ± 46 mg/L and conductivity values of 568 ± 47 μS/cm in groundwater at PZ1, the proportion of seawater in groundwater can reach over 50% near the sea. Analyses and measurements concern the entire water column due to observation well screening over almost the entire thickness of the aquifer.

### 3.1.5. STWW Infiltrated in the SAT

STWW is discharged successively into the three infiltration SAT ponds 1 to 3 according to a set schedule, but unfortunately, the switching dates from one pond to the next are not rigorously archived. Volumes infiltrated in each pond from 2010 to 2021 are thus not precisely known, but the provisional timetable and information gathered from the operator's technicians reconstructed a chronicle of the probable distribution of volumes by infiltration pond:

- From April to May in pond 1;
- From June to September in pond 2;
- From October to March in pond 3.

During winter, all three infiltration ponds may be filled to avoid overloading the hydraulic network and flooding the WWTP, without exact knowledge of the distribution of volumes between the three ponds. From January 2019 to June 2021, the rotation of STWW infiltration into the ponds was stopped for technical reasons. During that interval, the total STWW volume was continuously infiltrated into the three ponds without exact recordings of the distribution of volumes per pond. From June 2021, the rotation was restored according to the provisional planning. During the "winter" period from October to March, the average flow (inlet and outlet of the WWTP) was 1728 ± 845 m³/d with maxima of up to 5500 m³/d and minima of 379 m³/d. During "summer", from April to September, the average flow was lower, 1235 ± 299 m³/d with maxima of 3600 m³/d and minima of 174 m³/d. This variability is mainly explained by the infiltration of parasitic clear water into the sewer network, which is more significant during heavy rainfall and in

winter. In summer, about 500 m$^3$/d of the STWW is abstracted and stored for the irrigation of a nearby golf course.

Cl$^-$ is higher in the STWW (417 ± 234 mg/L) than in groundwater with little or no STWW (PZ1, Cl = 36 ± 46 mg/L). The Cl variations observed in wells near the NP1, AQ1-NP2, and AQ2-FRE4 infiltration ponds, follow a seasonal pattern, possibly linked to dilution by parasitic clear water in winter. Cl$^-$ is considered to be a conservative tracer of STWW in areas not affected by the sea. Nevertheless, only a few point analyses of Cl$^-$ are available for STWW, and only since 2016, not capturing the seasonal dynamics of Cl$^-$ concentrations in STWW.

The dilution variations in the STWW are, however, visible in the BOD5 concentration measurements acquired continuously since 2010; thus, Cl$^-$ concentrations in STWW could be estimated from these concentrations. The BOD5 concentration reaches maximum values in summer (July average 367 ± 77 mg/L) and minima in winter (February average, 115 ± 70 mg/L). BOD5 represents the input of anthropogenic organic matter. Assuming that the BOD5 variations are mainly due to the dilution of STWW water by parasitic clear water, they are (similarly to Cl$^-$) determined by the dilution occurring during winter. This is confirmed by a linear relationship between the Cl$^-$ and BOD5 concentrations (coefficient of determination of 0.65 on nine concentrations of Cl$^-$ measured). The Cl$^-$ time series can thus be reconstructed from the BOD5 concentration chronicle (371 measurements between 2006 and 2021), by considering a factor of Cl$^-$ = DBO5/0.61 between BOD5 and Cl$^-$ at the same dates. The Cl$^-$ concentration of STWW is then estimated to average 408 ± 200 mg/L.

### 3.1.6. Synthesis of the Conceptual Model

All flow to and within groundwater of the SAT site at Agon-Coutainville is summarised in Figure 4. The dune aquifer formed by homogeneous sand, 5 to 9 m thick, contains a free groundwater table that mainly flows horizontally to the west (gradient around 10$^{-3}$), the unsaturated zone being thin, less than 1.5 m. The maximum storage capacity in the unsaturated zone is about 0.495 to 0.990 Mm$^3$ over the total surface area of the aquifer. Recharge and discharge of the groundwater vary according to four potential conditions: (1) seasonal inflow of natural recharge with an average direct recharge of 1864 m$^3$/d and indirect recharge of 1579 m$^3$/d, with Cl$^-$ considered negligible; (2) exchanges with seawater that cause significant variations in potentiometric levels (e.g., AQ3-NP3, AQ4, and AQ5), as well as saline intrusion (near Blainville harbour and the coastline); (3) STWW inflow into the infiltration ponds with an average of 1525 m$^3$/d and 408 mg/L in Cl$^-$ concentrations, which varies seasonally in quality and quantity, depending on the contribution of parasitic clear water that increases in winter; and (4) the Goulot stream can drain groundwater or add water to the aquifer.

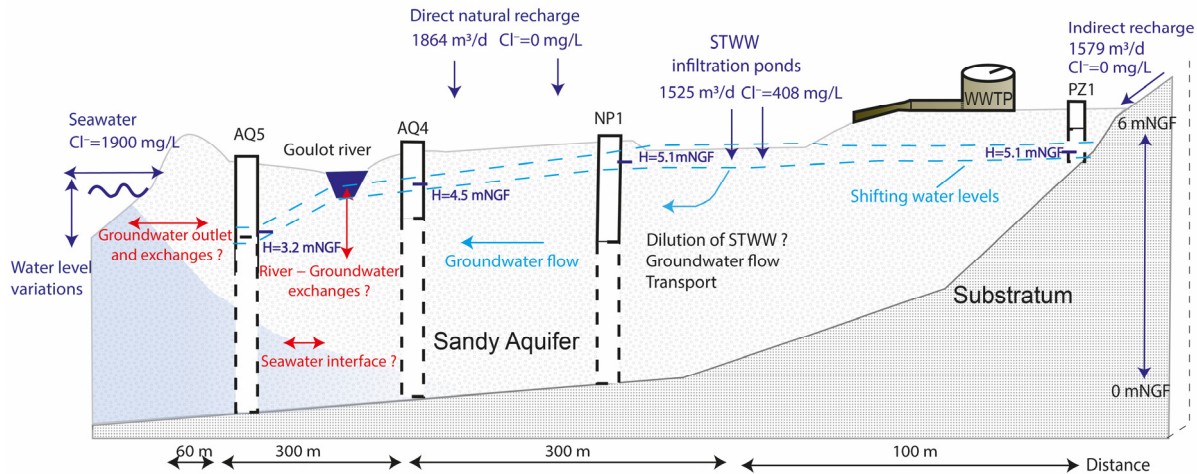

**Figure 4.** Conceptual model of the Agon-Coutainville SAT. Water balance and Cl$^-$ concentrations in the hydrosystem.

Considering the last point, there is considerable uncertainty about the stream's capacity of exchanging with the groundwater body. The variations in elevation of the water level in the stream, and the flow rates and hydraulic properties of the streambed, are unknown factors. Considering point (2), exchanges with the sea are not quantified, because they depend on hydrodynamic conditions of the groundwater and the tidal conditions.

*3.2. Flow and Solute Transport Equations*

In this study, groundwater flow and solute transport are governed by the differential equations of flow and non-reactive solute transport (for Cl). These equations can be solved, especially for multidimensional spaces, by numerical methods (finite elements, finite differences, and finite volumes).

The law of mass conservation, associated with Darcy's law, leads to the free sheet equation:

$$\text{div}(K.\text{grad}(H)) + Q = \frac{1}{\Delta z} S_L \frac{\partial H}{\partial t} \tag{2}$$

where K is the hydraulic conductivity [L·T$^{-1}$], H is the hydraulic head [L], Q is the external flow rate per unit area [L·T$^{-1}$], $\Delta z$ is the vertical thickness of layer [L], $S_L$ is the free sheet storage coefficient, equivalent to porosity [-], and *t* is the time [T]. The parameters K and $S_L$ affect not only the rate at which groundwater moves through the aquifer, but also the volume of water stored and the response of groundwater to stress in the aquifer system.

The 3D solute mass conservation equation, in the absence of interaction between solute and the solid phase and of biochemical transformation, is written as:

$$\frac{\partial(\theta_m \cdot C)}{\partial t} = div(\overline{\overline{D}} \, \theta_m \, \overrightarrow{grad}(C) - \overrightarrow{q}C) + q_m \tag{3}$$

where C is the concentration in mobile water [M·L$^{-3}$]; $\theta_m$ is the mobile water content [-]; D is the coefficient of dispersion [L$^2$·T$^{-1}$] with $\overline{\overline{D}}$ as the dispersion tensor comprising $D_L = \alpha_L |v|$ and $D_T = \alpha_T |v|$ with *v* as the pore water velocity defined by $v = q/n_c$, where $n_c$ is the effective porosity [L·T$^{-1}$]; $\alpha_L$ is the longitudinal dispersivity and $\alpha_T$ is the transverse dispersivity [L]; $q_m$ = Injected mass flux per unit volume [M·L$^{-3}$·T$^{-1}$]; t is the time [T]; and $\overrightarrow{q}$ is the Darcy velocity [L·T$^{-1}$]. Diffusion is assumed to be negligible compared with convection and dispersion.

For solving the flow and transport equations, the MARTHE computational code was used [31–33]. The differential equations of fluid flow and mass and energy transfer in three-dimensional porous media are solved numerically in a transient regime. The hydrodynamic calculation uses a finite-volume method (finite difference integration). For the transient mass transport, the TVD (total variation diminishing) method with a flow limiter is used for solving the equation system. Sub-time steps are automatically generated for the transport calculation by MARTHE, with respect to the low current conditions and thus reduce the numerical dispersion.

The convergence of iterative calculations is controlled by several criteria, mainly the average and maximum hydraulic head and mass differences between two successive iterations (globally on the full model). In practice, the state of convergence of a model is mainly evaluated by indicators concerning the hydraulic and mass balance for the whole model. Hydroclimatic, hydrological, and hydrogeological processes are coupled to model flow and solute transport. The direct natural recharge of the dune aquifer is calculated from the hydroclimatic calculation performed by the GARDENIA reservoir calculation scheme, coupled with the MARTHE aquifer-flow model. The RIVER module of the MARTHE code can calculate the groundwater/river exchanges.

Concerning such exchanges, for a river resting on a free water table, but whose bottom is clogged by a low-permeability mud layer, the exchanges between the river and groundwater table ($Q_{Éch}$ [L$^3$·T$^{-1}$]) are calculated according to Darcy's law, based on the

water level in the river in relation to the groundwater table level, and on the permeability of the river bed $K_R$ [L·T$^{-1}$]:

$$Q_{\acute{E}ch} = SURF_{\acute{E}ch} \cdot K_R \cdot \frac{(H_R - H_N)}{\acute{E}pais} \qquad (4)$$

where $H_R$ is the river level [L] and $H_N$ is the water table level [L], $SURF_{\acute{E}ch}$ is the river exchange area [L$^2$], and É*pais* is the thickness of the riverbed clogging [L].

The exchange between water table and river is calculated according to Equation (4). The volumes exchanged are limited by the calculated upstream river flow. The calculated runoff and overflow volumes (of the water table relative to the ground level) are taken over by the RIVER module and assigned to the river meshes.

### 3.3. Numerical Modelling: The Local SAT System in Its Hydrodynamic Aquifer Context

#### 3.3.1. Geometry

The modelled domain of the dune aquifer is shown on Figure 2. The distance between its northern (including Blainville harbour) and southern boundaries (including the Goulot river) is considered sufficient for reproducing the main regional flows in the SAT zone of influence. The aquifer top elevation corresponds to the topography derived from the DTM. In the modelled area, the elevation varies between 3.6 m.a.s.l. and 14.2 m.a.s.l. The bedrock elevation is derived from interpolation of the lithological sections observed in the wells. The aquifer is between 14.4 and 1.7 m thick from west to east.

#### 3.3.2. Discretisation in Space and Time

The domain is discretised into 22,192 square active meshes of 10 by 10 m. Due to thin nature of the unsaturated zone, and considering the predominantly horizontal flow, we chose a 2D horizontal single-layer model.

In total, 310 river meshes enable modelling of the Goulot stream, whose location is defined according to the drainage network of BDTOPO®. The width of the river sections is 2 m. Bed elevation is set at 0.5 m below the local topographic elevation and the clogged river bottom is 0.2 m thick.

In order to consider the temporal evolution of the hydrogeological system, the model was built in a transient regime, simultaneously integrating the water level variations, and the concentration and flow related to (1) tides, (2) natural recharge, and (3) the STWW infiltrated in the SAT. A daily time step was chosen to reproduce the variations related to these different forcings. The initial state of the Cl$^-$ concentration was unknown at the scale of the modelled domain, and a preliminary simulation covered the 2010 to 2017 period defining a steady state of groundwater concentrations from the different Cl$^-$ inputs from recharge, seawater, and the infiltrated STWW. The period used for calibration was 1 March 2017 to 31 December 2021, discretised in 4350 daily time steps.

#### 3.3.3. Boundary Conditions

The boundary conditions considered in the numerical model are: (1) overflow of the water table, (2) tidal boundary conditions (in the harbour and at the western limit of the modelled area, Figure 5), (3) direct natural recharge of the dune aquifer, and (4) indirect natural recharge at the eastern boundary of the model domain from basement catchments, the three STWW infiltration ponds, and the Goulot stream (Figure 5).

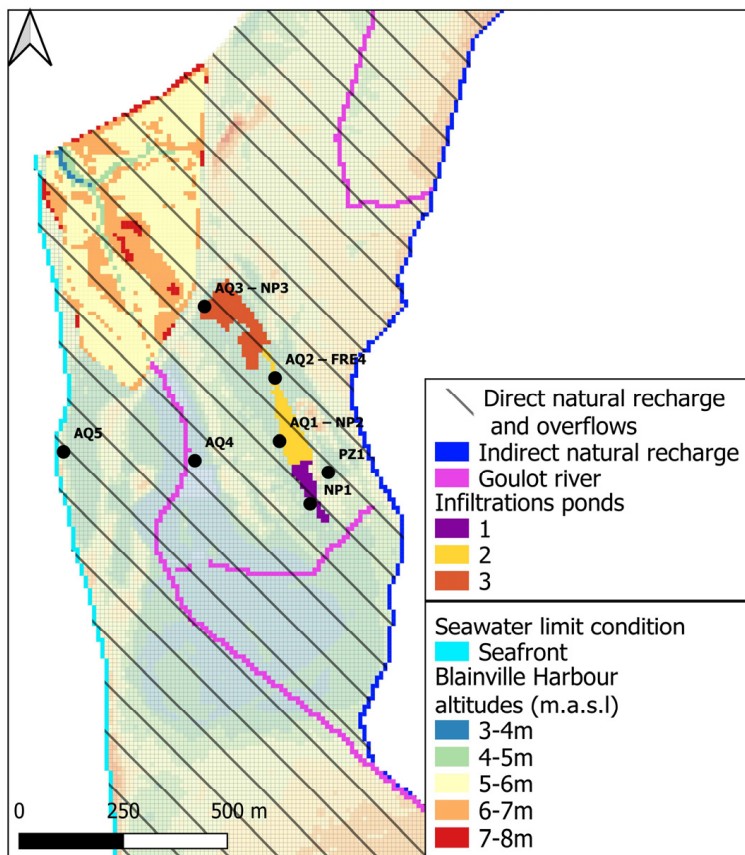

**Figure 5.** Boundary conditions applied to the numerical mesh hydrodynamic model MARTHE.

In each cell, the water table was considered to be free to overflow if the water table elevation exceeded the topographic elevation.

Direct natural recharge and runoff to the dune aquifer were calculated from the daily recorded rainfall and PTE time series from 2010 to 2021, via the GARDENIA calculation scheme. The calculation of direct natural recharge and associated low runoff was coupled to the MARTHE model and applied in each grid cell of the modelled area.

Indirect natural recharge flow to the dune aquifer from the basement watersheds was imposed on the aquifer meshes along the rim, continuously and consistently at 1500 $m^3/d$, based on estimates previously defined by the conceptual model. These fixed flows entering the edge meshes were homogeneously distributed, corresponding to a $Cl^-$ free flow of 6.0 $m^3/d$ per mesh.

Exchanges between water table and river were calculated from the water level fixed in each river mesh (Figure 5), constant and equal to the topographic elevation of the model meshes (or dune aquifer surface elevation). The riverbed elevation was set at −0.5 m from the river water table level.

The time-varying hydraulic head observed at AQ5 was fixed as a boundary condition on the meshes located at the western seafront boundary of the model (Figure 5). The measurements available at AQ5 over a one-year period between April 2017 and May 2018, and their potentiometric dynamics, were reconstructed for the modelling period from 2010 to 2021, based on a diffusivity equation via the computational code CATHERINE ([34], Supplementary Materials S2) that relies on the diffusivity equation and temporal sea level rise data reconstructed from the FES2014 harmonic component database ([35], Supplementary Materials S2).

The progression of the sea in Blainville harbour is shown by the imposed hydraulic-head boundary conditions for some of the meshes representing the harbour. The assignment of boundary conditions in these meshes varies in space as a function of sea level (Figure 5). At each computational time step, a hydraulic head corresponding to the maximum daily

sea level was fixed on the aquifer meshes in the harbour with a topographic elevation below that level. At the highest tide levels (6 to 7 m.a.s.l.), 96% of the harbour is considered as a fixed hydraulic head, while at a sea-surface elevation below 3 m.a.s.l., there are no fixed heads in the harbour.

These boundary conditions imply lateral water exchange in the dune aquifer in both directions. Water entering the aquifer from these boundaries has a Cl⁻ concentration set to that of the sea at 19,000 mg/L [30].

The daily STWW flows imposed on the model vary over time, based on data acquired by the operator (from 2010 to 2021) by subtracting summer withdrawals for the water needs of the golf course. Daily flow was imposed in a homogeneous manner on the meshes of each pond, impounded according to the provisional schedule of alternating pond supply (Figure 6) as identified by the conceptual model. For periods when the gates are open without precise knowledge of the discharge position (Figure 6) from December to February (before January 2019), the distribution of STWW flow was 16%, 34%, and 49% for ponds 1, 2, and 3, respectively. After the alternation was halted (January 2019 to June 2021), the flow distribution was set at 60%, 40%, and 20% for ponds 1, 2 and 3, respectively. Outside these periods, the meshes of the non-fed ponds were subjected to the flow and transport of solutes without STWW injection constraints.

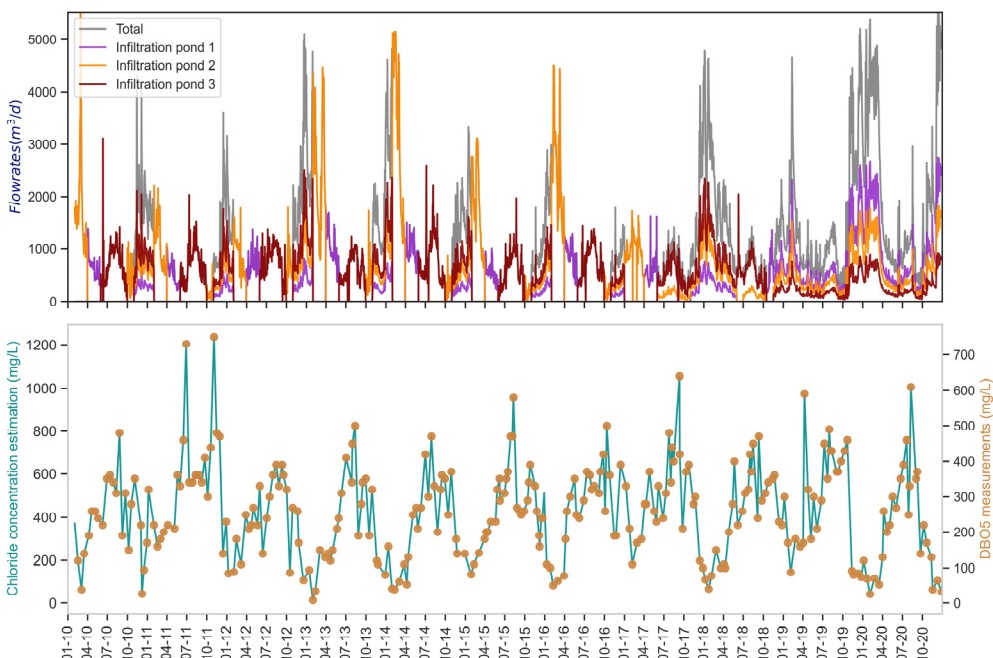

**Figure 6.** Time series of STWW infiltration flows in the different infiltration ponds from 2010 to 2021 and Cl⁻ concentrations in the STWW reconstructed from the measured BOD5 concentrations.

The Cl⁻ concentrations of STWW infiltrated in the ponds were variable over time, the time series imposed in the model being reconstructed (Figure 6) from the chronicles of BOD5/0.61 concentration, as explained above.

### 3.3.4. Initial Conditions

The initial conditions of hydraulic heads and Cl⁻ concentrations of the water table as of 1 March 2017, cannot be defined in all model grids. These were calculated with a seven-year model calculation, considering the conditions of tides, STWW flow, and natural recharge since 2010. It was ensured that both water table level and Cl⁻ concentrations simulated at the end of this model equilibration period were of the same order of magnitude as those measured in the observation wells.

### 3.3.5. Model Calibration

The model was calibrated for the period 1 March 2017 to 31 December 2021 by error testing over a range of parameter values that realistically represented the aquifer characteristics, as identified by the conceptual model. Adjustment of the hydrodynamic and transport parameters reproduced the dynamics of the observed water levels and, as far as possible, those of the $Cl^-$ concentrations observed in the wells. Stream flow orders of magnitude were taken from the literature in the absence of measurements. The fitted hydrodynamic parameters were the hydraulic conductivity (K) and a free-sheet storage coefficient ($S_L$), assigned to two zones corresponding to outcrops of dune sand and aeolian sand. Locally, the permeability and clogging thickness of the river ($K_R$, *Épais*) were adjusted as well.

The solute transport parameters in the aquifer, effective porosity, and dispersivity were calibrated. As the effective porosity corresponds to the free storage coefficient $S_L$, these two values are equal. The longitudinal dispersivity value, $\alpha_L$, varies with the scale of investigation and reflects the influence of variability in aquifer heterogeneities. For sandy aquifers, $\alpha_L$ values between 10 and 100 m are acceptable at our scale of investigation [36,37]. A transverse dispersivity $\alpha_T$ weaker than $\alpha_L$ [26] was set here at $\alpha_L/10$.

### 3.3.6. Model Sensitivity

Different pairs of hydrodynamic and hydrodispersive parameters were tested to estimate the model sensitivity. Such analyses concern: (1) hydraulic conductivity values of the aquifer tested over the possible range of coarse sand from $2.0 \times 10^{-3}$ m·s$^{-1}$ to $2.0 \times 10^{-4}$ m·s$^{-1}$; (2) a homogeneous hydraulic conductivity value over the entire aquifer; (3) porosity values, free storage porosity values, and free storage coefficients tested over the ranges of the bibliography for coarse sand (20 to 35%); and (4) a longitudinal dispersivity parameter tested here over a range of 10 to 100 m.

For direct natural recharge of the dune aquifer, the GARDENIA model parameters were modified to simulate the effect of about 10% less recharge. For indirect natural recharge, two simulations were tested: (1) The indirect recharge time series calculated by the GARDENIA model was integrated into the numerical model; the flow calculated by GARDENIA (1579 m$^3$/d) was similar to the average flow initially set at 1500 m$^3$/d, but varied with lower flow in summer and higher flow in periods of high precipitation. (2) A hydraulic head corresponding to the minimum value of the levels observed at PZ1 (4.5 m$^3$/d) at PZ1 (4.5 m.a.s.l.) was imposed at the boundary of the dune aquifer.

In a simulation, only the tidal boundary conditions on the coastline west of the model were maintained (according to the potentiometric levels reconstructed at AQ5 and a Cl concentration of 19,000 mg/L) without considering flooding of the harbour.

In view of the uncertainties on the Goulot water levels and geometry, these were modified according to two conditions: (1) the water level was lowered by 0.4 m with respect to the topographic elevation in the downstream part of the stream (between the infiltration ponds and the sea); and (2) the riverbed level was raised from 0.5 to 1.5 m.a.s.l.

## 4. Results

### 4.1. Calibrated Hydrodynamic and Hydrodispersive Parameters

The hydrodynamic and hydrodispersive parameters were calibrated over the period 3 January 2017 to 31 December 2021 for two geological areas, the recent dunes and the aeolian sands (Figure 1). In these two zones, the values of the free storage coefficients and porosity are 20% and 10%, respectively, and those of hydraulic conductivity are $2.0 \times 10^{-3}$ m·s$^{-1}$ and $5.0 \times 10^{-6}$ m·s$^{-1}$, respectively. The dispersivity ($\alpha_L$ = 10 mm and $\alpha_T$ = 1 m) and groundwater/river exchange parameters ($K_r = 1 \times 10^{-6}$ m·s$^{-1}$) were kept uniform over the modelled area in the absence of field observations indicating possible heterogeneities.

Observed and simulated mean hydraulic heads show a 1:1 correlation, indicating a robust calibration of the hydrodynamic parameters, although with nuances highlighted by the root mean square error results (RMSEs, Figure 7). For example, the lowest RMSEs were

calculated for wells PTC6 and AQ4, with 0.10 m and 0.12 m, respectively. Higher RMSE of 0.17 m, 0.18 m, 0.29 m, and 0.24 m correspond to wells located near the infiltration ponds (AQ1-NP2, NP1, and AQ2-FRE4) and near the coast (AQ5). Finally, the maximum RMSE values were calculated at 0.31 m and 0.35 m for wells PZ1 and AQ3-NP3.

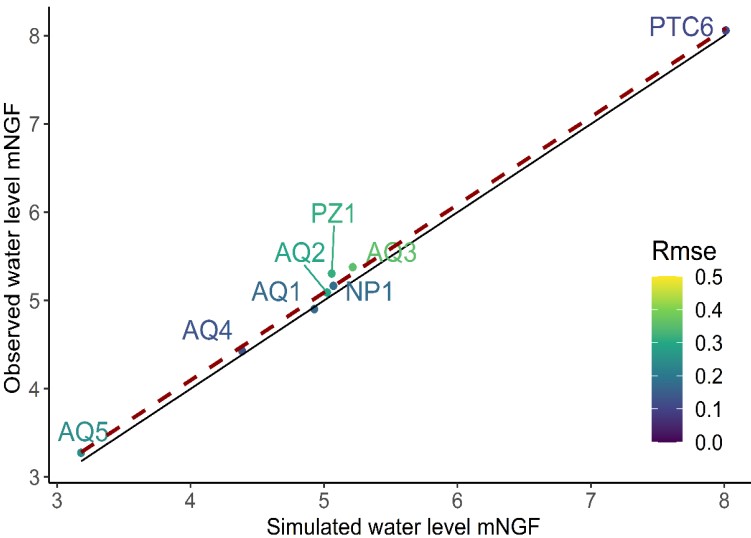

**Figure 7.** Plot of observed and simulated mean water levels and associated calculated RMSE for the different observation wells. The black line is an x = y function, the red dashed line is the calculated fit line.

In addition, the modelled hydraulic heads reproduce the dynamics observed in the potentiometric chronicles of the annual high- and low-water cycles (Figure 8). The tidal cycles from high to low tides (14.8-day period) and their visible effects on the water-level measurements are reproduced in wells AQ4, AQ3-NP3, and AQ5.

Occasional differences—related to a lack of data on which ponds were activated—are identified in the observation wells close to the infiltration ponds (e.g., August–October 2021 at NP1 and PZ1), or are continuous as at AQ3-NP3 where the model overestimates the impact of high- and low-tide cycles.

The calibration quality can also be appreciated from $Cl^-$ concentrations. Over the calibration period, the simulated concentrations respect the measured orders of magnitude (Figure 8). Concentration variations are generally well reproduced by the model, although with a strong overestimation of concentrations near Blainville harbour (AQ3-NP3), where average differences of 6700 mg/L are caused by the nearby marine conditions. Upstream, near the east edge (PZ1), $Cl^-$ is slightly overestimated (on average by 66 mg/L), related to the indirect natural recharge boundary condition. In the observation wells close to infiltration ponds, the differences between observed and simulated average concentrations vary between 12 mg/L (AQ1-NP2) and 146 mg/L (AQ2-FRE4) and, locally, 500 mg/L (overestimation or underestimation). The calculated river flow varies from 8600 $m^3$/d to 25,000 $m^3$/d at the outlet of the Goulot stream. The calculated flow is higher than the expected flow (500–5000 $m^3$/d).

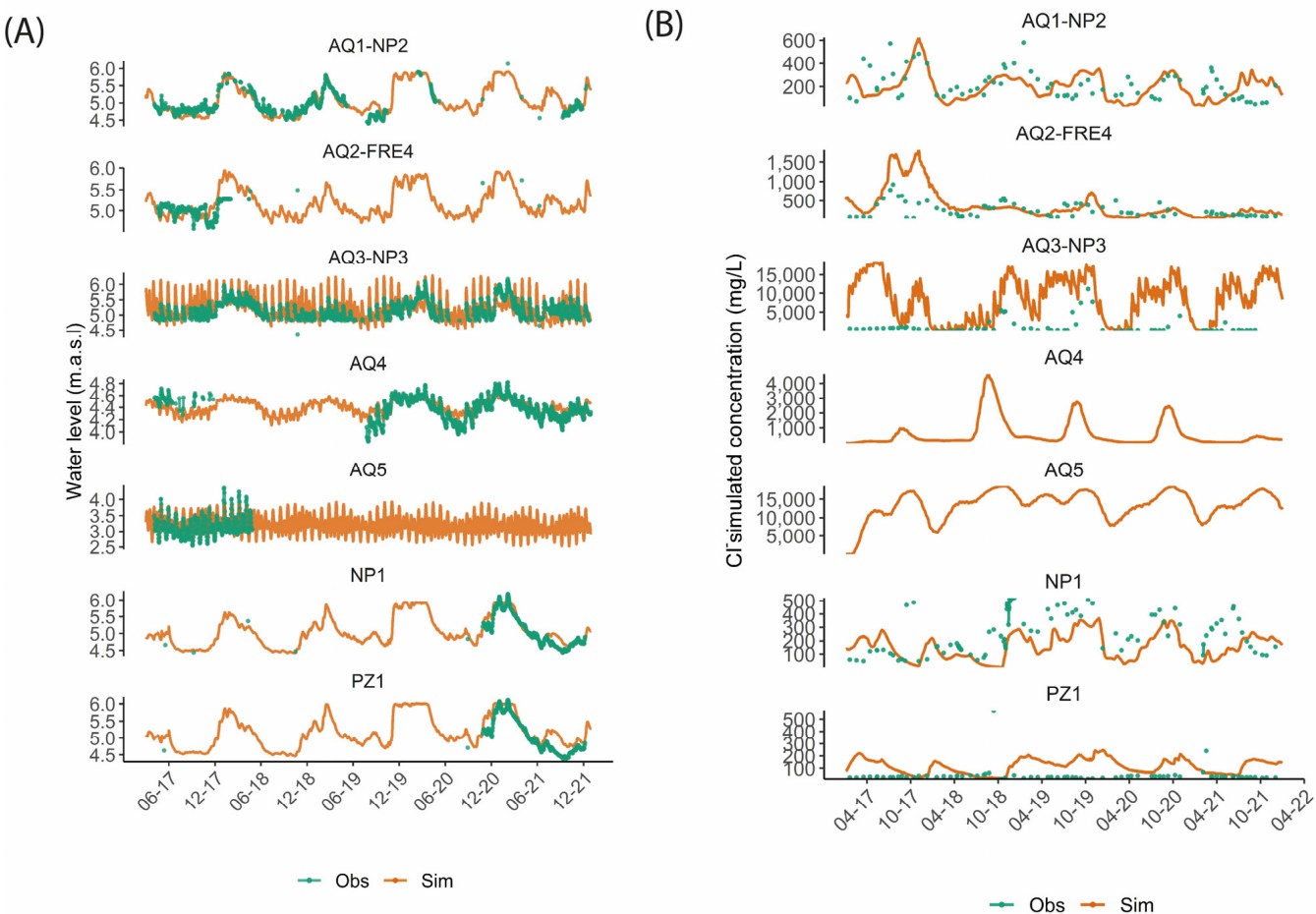

**Figure 8.** Time series of observed and simulated (**A**) piezometric levels and (**B**) Cl⁻ concentration over the period 2017 to 2021 at piezometers AQ1-NP2, AQ2-FRE4, AQ3-NP3, AQ4, AQ5, NP1, and PZ1.

*4.2. Hydrodynamic Balances in 2017–2021*

The hydrodynamic balances for the period 2017 to 2021 show the water exchanges between the SAT zone and its hydrodynamic environment. The area for the balance calculation (Figure 9) of 0.6 km² is delimited by bedrock outcrops and the coastline, excluding Blainville harbour to the north and part of the Goulot upstream pond to the south.

The amounts of water entering the aquifer in the SAT zone (Figure 9) are, on average, 34% STWW (0.57 Mm³), 23% from groundwater-to-river exchanges (0.39 Mm³), 23% natural recharge (12% or 0.20 Mm³ direct recharge and 11% or 0.18 Mm³ indirect recharge), and 18% from outside the study area, mainly from Blainville harbour (0.3 Mm³). The different water inputs into the aquifer, from streams, STWW, natural recharge, and the sea (harbour and coastal areas) result in water mixtures of variable chemical composition.

On average, 54% of groundwater outflow occurs through fixed hydraulic head boundaries to the sea (0.9 Mm³) and 45% through groundwater overflow (0.7 Mm³).

Natural recharge (0.28 Mm³) and STWW flow (0.37 Mm³) were lowest in hydrogeological year 2018–2019, compared with the "wet" year 2020–2021 with 0.43 Mm³ and 0.71 Mm³, respectively. Given the low storage capacity of the aquifer, such "dry" and "wet" years have the same impact on overflow and the volumes leaving the outlet towards the sea.

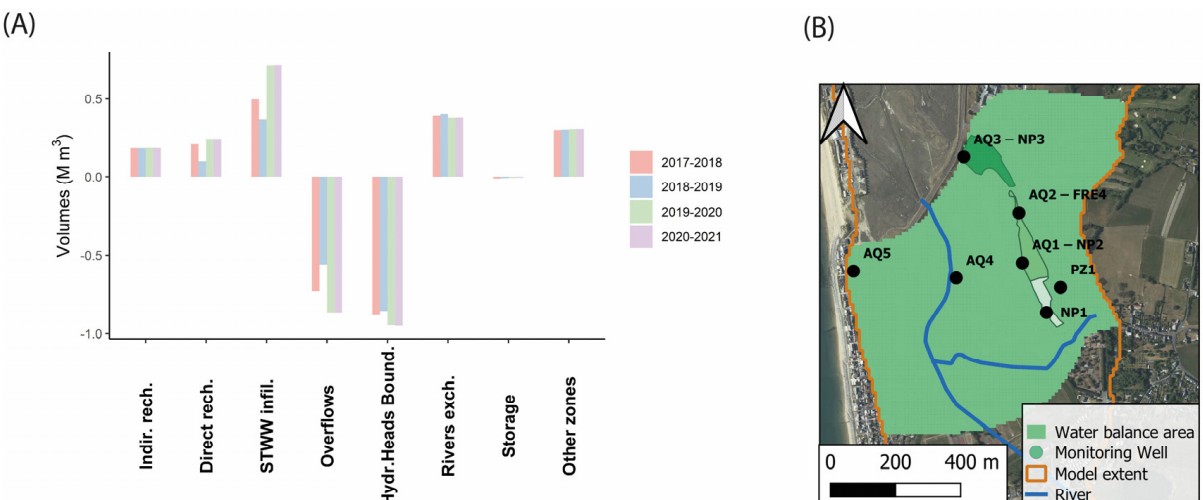

**Figure 9.** (**A**) Hydrodynamic balances by hydrogeologic year (e.g., 2017–2018: 1 October 2017–1 October 2018) on direct and indirect natural recharge, infiltrated STWW, overflow, fixed hydraulic heads at the western boundary of the model (marine conditions), aquifer/river exchanges, aquifer storage, and volumes from other zones (balance area is shown in (**B**). Negative values indicate water outflow from the aquifer and positive values indicate water inflow into the aquifer.

### 4.3. Calculated Piezometric Maps and Main Flow Lines

The simulated potentiometric maps for May 2019 and May 2021 (Figure 10) show the high-water periods after poor winter recharge in 2018–2019 and heavy winter recharge in 2020–2021. The hydraulic heads on 1 May 2019, and 2021 varied between a minimum of 2.96 m.a.s.l. west of the model and a maximum of 14.16 m.a.s.l. to the east. The main flow directions were NE–SW from the infiltration ponds to the sea (east-to-west), obliquely crossing the downstream part of the Goulot stream and bending towards the sea perpendicular to the coastline. The flow directions at three different times (December, May, and August of both 2018–2019 and 2020–2021, Figure 10) showed little change over the course of a hydrogeological year, except between the northernmost seepage pond 3 and the Goulot stream, where tidal conditions in the harbour play a significant role (e.g., May 2020, Figure 10).

### 4.4. Spatialised Variations in Flow Velocities and Proportions of STWW

The mean values with coefficients of variation (CV) of flow velocities and STWW proportions are shown spatially for two extreme hydrogeological situations; in a low-rainfall hydrogeological year (2018–2019) and a high-rainfall year (2020–2021). The averages and CVs were calculated from real velocities calculated over two time steps per month (i.e., 24 time steps per year) by the numerical model.

Variations in flow velocities and STWW proportions in groundwater were analysed over four segments (Figure 11, numbered '1' with a length of 265 m, '2' with 325 m, '3' with 250 m, and '4' with 190 m). These are defined by the main flow lines from: (1) the infiltration ponds to the Goulot stream, the area with the highest STWW proportions (one line per pond); and (2) from the Goulot stream to the coast, an area with lower STWW proportions. Variations were calculated over the 2017–2021 period on a bi-monthly basis for each segment by averaging the simulated values across the segment meshes.

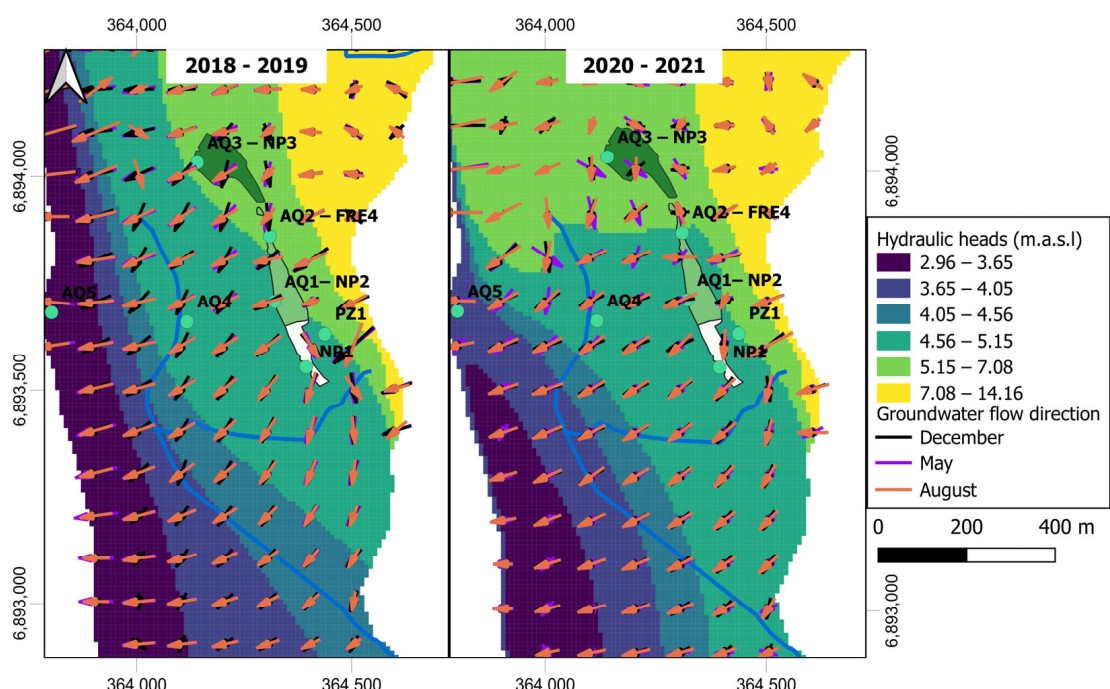

**Figure 10.** Calculated hydraulic heads on 1 May 2019, and 1 May 2021. The arrows show groundwater flow directions for three periods: December (black), May (purple), and August (orange).

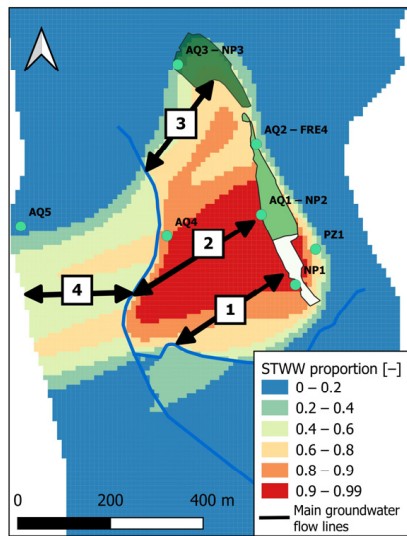

**Figure 11.** Selection of the main groundwater flow lines for calculating the average velocities between infiltration ponds 1, 2, and 3 and the Goulot, and 4 between the Goulot and the sea.

The average flow velocities calculated over all meshes in segments 1, 2, 3, and 4 were 2.5 m/d, 2.7 m/d, 3.0 m/d, and 3.7 m/d, respectively (Figure 12), for the period 2017–2021. The highest velocities were calculated in winter, in February, March, and April (4.6 m/d, 4.7 m/d, 5.5 m/d, and 6.7 m/d, respectively). The values were minimal during summer, August to October, (1.0 m/d, 1.3 m/d, 1.7 m/d, and 0.9 m/d) indicating a strong seasonality. The amplitude of the velocity variations was exacerbated on flow lines 3 and 4 near the model boundary. Marine tidal cycles induced other, weaker, variations in about 0.50 m/d.

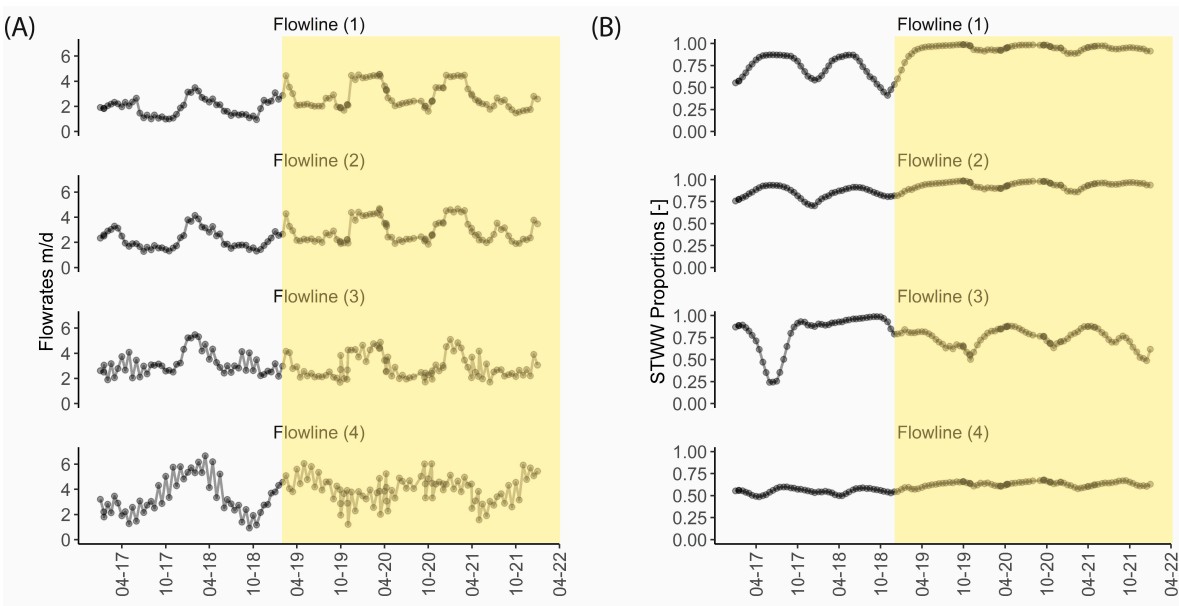

**Figure 12.** Variations in (**A**) average flow velocities and (**B**) STWW proportions, calculated by the model according to the flow lines from the infiltration ponds 1, 2, and 3 (Flowlines 1, 2, and 3) and between the Goulot and the sea (Flowline 4). The orange area represents the period when the alternation of STWW discharges into the different infiltration ponds was interrupted.

The mean residence times calculated from flow velocity results and the length of the segments of lines 1, 2, 3, and 4 (Table 2) are 107 d, 122 d, 82 d, and 50 d, respectively. In winter, the mean residence times are shorter with minimum values of 28 to 69 days with the lowest values remaining on lines 3 and 4 near the sea. However, under summer conditions they are longer, with maximum values of 148 d to 282 d, again not counting any marine influence.

**Table 2.** Descriptive statistics of the average velocities calculated according to the main flow lines and associated residence times.

| Name<br>Length (m) | Flowline 1<br>265 | Flowline 2<br>325 | Flowline 3<br>250 | Flowline 3<br>190 |
|---|---|---|---|---|
| Velocity (m/d) | | | | |
| Mean | 2.48 | 2.67 | 3.03 | 3.77 |
| Min | 0.94 | 1.28 | 1.69 | 0.92 |
| Max | 4.55 | 4.69 | 5.48 | 6.67 |
| Residence time (d) | | | | |
| Mean | 107 | 122 | 82 | 50 |
| Min | 58 | 69 | 46 | 28 |
| Max | 282 | 254 | 148 | 207 |

The average proportions of STWW mixing in the aquifer, calculated on flowlines 1 and 2 over the period 2017 to 2021, are, on average, 86% and 91%, and vary significantly when the alternating discharge from one pond to the next was effective, before January 2019, Figure 12). During this period, the proportions calculated varied from 45% to 86% for line 1 and from 75% to 93% for line 2. However, after interruption of the alternate water releases, the average STWW proportions on lines 1 and 2 reached plateaus of 86% and 97%, respectively, and were only occasionally lowered by 10%. On flowline 3, the STWW proportions vary more strongly, with an average of 77%, a minimum of 24% in May 2017, and a maximum of 98% in January 2019. Large variations in proportions persist, such as

50% in November 2019. The STWW proportions for line 4 between the coast and the Goulot are relatively stable, with an average of 60% regardless of the infiltration ponds used.

## 5. Discussion

The numerical flow and transport model calculates, over time and at any point of the Agon-Coutainville aquifer, the flow velocities and proportion of STWW in groundwater. Of particular interest are the trajectories followed by STWW infiltrated through the three ponds. This information, unknown until now, is a key parameter for interpreting the reactivity of TrOCs measured in the field. For this purpose, the developed model was based on: (1) the major flow and mixing processes integrating the dynamics of natural and anthropogenic forcings, and (2) a calibration of the parameters with regard to the available data. An iterative approach with novel field measurements and continued modelling would further improve our understanding of this coastal hydrosystem.

### 5.1. Interactions between SAT and Surrounding Natural and Anthropic Dynamics

At the aquifer scale, the STWW infiltrated in the three ponds flows toward the coast, locally intersected perpendicularly by the drainage network. Over two hydrological years with distinct recharge conditions (2017–2018 and 2020–2021, Figure 12), the results show: (1) the preservation of a high proportion of STWW (60% to 99%) between the infiltration ponds and the stream, and (2) flow velocities from 2 to 5 m/d between the infiltration ponds and the stream.

However, intra-annual calculations show that the SAT is subject to strong seasonal modifications of velocities and dilutions of STWW. Locally, the velocities (along the main flowlines) vary from 0.9 to 5.5 m/d from the infiltration ponds to the coast (equivalent to residence times of 74 and 489 days, respectively) and episodes of plume dilution may occur depending upon the influence of natural and anthropogenic forcings (Figure 11). Particular behaviour is observed for specific zones, from infiltration ponds 1 and 2—farther from the coast—to the stream, from infiltration pond 3 to the coast, and from the stream to the coast.

From ponds 1 and 2 to the stream, the flow velocities show strong intra-annual seasonal variations. The general increase in potentiometric levels linked to natural winter recharge and accentuated by high infiltrated STWW volumes, correlated to precipitation, can double the flow velocities during winter compared with summer (4.7 m/d in winter to 2.6 m/d in summer). In this area with over 90% of STWW on average, the mean residence time of infiltrated STWW varies from 58 days in winter to 282 days during summer, without any influence of natural recharge on the dilution as long as the supply from the ponds is continuous (no alternation). When the ponds are fed alternately, natural recharge causes a significant dilution of the STWW plume in winter, of up to 50% in pond 1 and 25% in pond 2. Even when using the correlation between BOD5 measurements and $Cl^-$ concentrations for reconstructing the seasonal variations in $Cl^-$ concentrations in STWW, there is high confidence in the STWW velocities and proportions calculated from these flow directions, in view of the quality of the calibration measurements obtained near the infiltration ponds, both for the hydraulic heads and the $Cl^-$ concentrations.

Between pond 3 and the sea along Blainville harbour, the proportions of STWW in groundwater are of the same order of magnitude—on average, 77%—as those from ponds 1 and 2 with higher and more variable flow velocities. The variations in flow velocity—and the mean residence time between 46 and 148 days—between pond 3 and the stream are caused by all driving factors, i.e., the sea, STWW, and rainfall. The cyclicity of flow velocities is both seasonal (1.7 m/d minimum in summer up to 5 m/d in winter) and monthly, with a lesser amplitude linked to tidal cycles and spring tides. The occasional dilution of STWW plumes can reach 24%. These one-off events, particularly during the autumn equinoxes, are the result of a combination of (1) factors related to intense marine dynamics, (2) lower volumes of STWW supply to pond 3 due to operational conditions, and (3) higher groundwater levels during the winter recharge period. These major variations in both flow velocities and STWW proportions show that this specific area is strongly

influenced by the sea. It is certain that harbour flooding is a key process with a significant effect on the SAT during the equinox periods. Some uncertainties are related to flow processes not considered in the model: (1) the meanders of the harbour surface and their interaction with underlying groundwater; (2) meteorological effects on sea levels; and (3) effects related to changes in water density.

Between the stream and the sea, an increase in average flow velocities is caused by the stream contribution and its exchanges with groundwater (0.39 $Mm^3$/year of water contributed by the river to groundwater). Between the stream and the sea, the mean water residence time varies from 28 to 207 days, being very sensitive to seasonal dynamics and, to a lesser extent, to sea level variations; this leads to a mean residence time between 74 and 489 days between the infiltration ponds and the coast. Stream input diminishes the calculated STWW proportions to, on average, 60%. There are uncertainties in the calculation of dilution and flow velocities, due to a less-than-robust estimate of water-to-river exchanges induced by our only partial knowledge of the stream. Even if the sensitivity analysis (stream geometry) confirms that the stream induces an increase in velocity and high dilution in this area, it is not excluded that flow could be reversed when considering a finer description of the stream dynamics, thus modifying the volumes exchanged with groundwater. Additional investigations of flow, water levels, and streambed geometry would refine our understanding of exchange dynamics, for better estimating the role of the stream in the SAT dynamics.

The identified model limitations are related to two modelling assumptions: (1) a homogeneous aquifer with no stratification or diversity of geologic facies within the porous medium, and (2) negligible density effects in these coastal environments.

Both assumptions mainly refer to the transition zone (freshwater–saltwater interface), where the fluid density varies in time and space as a function of temperature and salt concentration. When these mechanisms are essential, variable density flow modelling helps explaining saltwater intrusion observations [38–40]. Considering such processes in the model applied to the Agon-Coutainville SAT would modify the simulated results, such as: (1) decreased flow velocities of saline groundwater relative to fresh groundwater; (2) vertical stratification of more dense saline groundwater relative to fresh groundwater; and (3) increased velocity of the infiltrating STWW freshwater plume due to the existence of this stratification. Concerning the STWW proportion, effects are also possible due to the density-driven progression of the STWW plume, which extends over a superficial part of the aquifer, less prone to seawater dilution. These effects are nevertheless assumed to be negligible in areas where no saltwater wedge is observed, notably close to infiltration ponds 1 and 2. In areas closer to the coast and the harbour (pond 3), additional investigations, such as the acquisition of an electrical-conductivity profile, or other geophysical methods such as electromagnetic induction [6], can be used for defining the saltwater wedge and its extent in the Agon-Coutainville site.

Sensitivity analysis shows that the results of flow velocities and STWW proportions are sensitive to permeability and porosity, and to the choices of conceptualising indirect recharge. Additional in situ measurements, such as pumping tests, and tracer tests (for evaluating soil porosity) can validate the choice of parameters for calibrating the hydrodynamic model; further measurements for identifying runoff upstream of the aquifer (such as flow measurements and the use of water isotope tracers) can also be suggested.

### 5.2. Effects of SAT Variations on the Fate of TrOCs at the Hydrosystem Scale

Under physico-chemical conditions favourable to degradation, most TrOCs are more degraded when the residence time is longer, but some persistent compounds are not degraded even over long residence times [12,41], although some 'persistent' compounds, such as carbamazepine, may be degraded under certain conditions (redox conditions or the availability of organic matter [42]). Assuming homogeneous reactivity to groundwater flow for biodegradable compounds with low refractoriness (e.g., a half-life of 1–10 days), there is little chance of finding these compounds at the outlet of our watershed due to the

long water-residence times in the SAT. For compounds that are refractory (e.g., 50 days), a residence time of 74 to 489 days between infiltration ponds and coast, as calculated for the Agon-Coutainville SAT scheme, would favour a concentration decrease of over 50%, amplified (by 40%), in the downstream parts of the aquifer, by dilution from the Goulot stream (assuming that this water does not contain TrOCs), and/or by seawater (saline intrusion and harbour flooding). For transformation products, higher residence times, associated with conditions enabling the degradation of molecules, would then conversely lead to an increase in their concentration within the aquifer [43].

TrOC reactivity in the Agon-Coutainville SAT has already been quantified [20], at the scale of an infiltration pond and at laboratory scale [44]. Our modelling results show that the reactivity observed at these scales can be modified over time and space due to modifications of groundwater flow velocity and dilution.

Between infiltration ponds 1 and 2 and the Goulot stream, a strong variability in SAT reactivity is expected due to the alternation of infiltration ponds and seasonal variations, and their effects on key factors that influence TrOC degradation [17].

Microbial activity in SAT environments receiving STWW is strongly related to the availability of dissolved biodegradable organic matter (DBOM) used as a co-substrate for the metabolic transformation of TrOCs [45–47]. Other studies [46,47] have shown that microbial adaptations to low-biodegradable dissolved organic carbon (BDOC) conditions can strongly increase the biodegradation of TrOCs. Alternating recharge of infiltration ponds and groundwater dilution can stimulate microbial diversity by decreasing available BDOC, and thus, possibly the transformation of TrOCs.

The redox state of groundwater is another key parameter that controls the degradation of TrOCs [17] and many other molecules [14,48]. Often, different redox conditions (such as oxic–penoxic–suboxic–anoxic zones [48,49]) are established in SAT systems, due to the high input of organic matter to the infiltration pond surface and equilibrium with the atmosphere, which induces a lower oxygen availability along the flow. Stopping the alternating infiltration between ponds would modify the redox zones installed along system flows, and thus, the degradation of redox-sensitive compounds. As shown by our model, seasonal variations strongly modify the SAT dynamics. From a reactive viewpoint, seasonal temperature variations can strongly influence TrOC degradation through lower microbial activity and the development dynamics of various redox zones in the soil and aquifer [49,50]. In winter, microbial activity can be reduced by lower temperatures [51]. Over the same period, the calculated mean residence times are shorter, contributing to a lower overall reactive efficiency of the SAT. Nevertheless, the TrOC concentrations in STWW are lower in winter (dilution by clear parasitic water), which reduces the impact of a lower system reactivity. The higher temperatures in summer, coupled with a higher input of biodegradable organic matter into the STWW, create more favourable conditions for TrOC degradation [51]. These conditions, combined with longer summer mean residence times, would increase the overall SAT efficiency over this period, compensating for the higher TrOC concentration in STWW.

A change in the aquifer capacity is also considered. Indeed, the sorption of many TrOCs is related to the proportion of organic matter [52], the hydrophobicity of the molecules [52,53], and the characteristics of organic matter [12,54–56]. Variation in dissolved organic matter and of the proportion of organic matter can therefore modify the mobility of TrOCs in the aquifer.

Close to the harbour (pond 3) and the sea, plumes of infiltrated STWW are subject to seawater intrusions that can influence TrOC reactivity. There is little information on TrOC degradation in marine or estuarine environments; one study, for example, showed that degradation rates in marine surface waters are lower than in freshwater [57]. However, in groundwater, further work is needed to verify the impact of a coastal environment on TrOC reactivity. When seawater moves into a coastal aquifer, the modification of the chemical gradient can strongly modify microbial activity, and thus, the degradation of TrOCs, as well as their mobility.

During their migration, the molecules will encounter areas with more efficient bacterial communities, boosting TrOC degradation in soil and aquifer. Such phenomena further evolve according to temperature, microbiological activity, and the presence of organic matter. In the Agon-Coutainville study site, or any other SAT site, a higher reactivity of TrOCs is expected near the ponds [58], even if the transformation of more refractory molecules can continue after longer travel times [46]. Estimates of degradation are not easily extrapolated to an entire watershed, or transposable to another site. At the scale of an SAT, the spatialisation and dynamics of TrOC reactivity, in perpetual feedback with those of water flows, must be considered for more precisely predicting the degradation of TrOCs in such systems. The empirical formalisms established in the current literature on their degradation and sorption provide an initial characterisation of such reactivity, but are not necessarily adapted to the aquifer scale [59,60].

Spatially, the mobility of TrOCs is also influenced by organic matter content, which is higher at the beginning of infiltration, thus causing higher sorption to the first soil horizons [52]. During runoff, TrOC mobility may be increased in areas with little organic matter, unless the presence of minerals (such as oxy-hydroxides or clays) in the aquifer will decrease the mobility of charged TrOCs [61].

## 6. Conclusions

Our numerical model was developed for monitoring trace organic compounds (TROCs) in the Agon-Coutainville coastal soil aquifer treatment (SAT) site for managed aquifer recharge. It assesses water flow velocities and the optimal proportions of secondary treated wastewater (STWW), from the infiltration ponds to the aquifer outlet, and at aquifer scale. Before, such information was only available on an ad hoc and local infiltration-pond-scale [20]. Flow velocities and STWW proportions in the aquifer can now be quantified over time at any point in the aquifer.

Our results show strong dynamics in the SAT functioning, including large variations (from 70 to 500 days) in STWW mean residence time between the infiltration pond to the outfall, which is mainly related to STWW discharge conditions and meteorological factors. The dilution rates of STWWs vary, depending on operational conditions (infiltration pond feeding), on proximity to the coast (mixing with saltwater) in the area near Blainville harbour, and on the inflow of Goulot river water. Strong SAT dynamics can modify—at hydrosystem and annual scales—the reactivity of TrOCs, obtained from a local-scale study, based on first-order degradation coefficients ($\mu$) and delay coefficients (R) [14,60,62].

Seasonal STWW concentration variations, mean residence times, and temperature differences, will modify SAT reactivity as a function of time. The precise quantification of residence time and dilutions is an initial step for the further interpretation of hydrogeo-chemical data, such as TrOC quantities in groundwater, or for further investigations or reactive modelling that consider modifications of reactivity within the SAT (changes in redox conditions, organic matter availability, etc.). Our simulation results show that the observed removal of TrOCs near infiltration ponds can only be explained by geochemical reactivity within the SAT when the ponds are in use. The seasonality of residence times, STWW concentrations, and reactivity conditions shows synergy dynamics, resulting in low concentrations at the aquifer outlet. In winter, reactivity conditions are less favourable for TrOC degradation, but lower concentrations are found in STWW than in summer.

Additional measurements of TrOCs, as well as of redox conditions, organic matter availability, temperature, etc., with further modelling support, will help interpret effective SAT reactivity, depending upon varying flow conditions and dynamic factors that influence such reactivity. This will help further understanding of the key conditions that influence TrOC degradation at the scale of an operational SAT site.

Finally, our hydrodynamic numerical model can be used for optimising the position of STWW in an infiltration pond, so as to improve the mean residence time, and thus, the degradation of organic trace molecules. Such improvement will be the result of a much

better understanding of tidal, water flow, and weather conditions at the Agon-Coutainville managed aquifer recharge site.

**Supplementary Materials:** The following supporting information can be downloaded at: https://www.mdpi.com/article/10.3390/w15050934/s1, Figure S1: Monthly inter-annual variations calculated over the period from 2006 to 2021 at the Gouville-Sur-Mer station; Figure S2: Complete GARDENIA hydro-climatic balance scheme; Table S1: Gardenia parameters applied in the calculation of natural recharge in Agon-Coutainville; Figure S3: Estimation of "direct" natural recharge on the dune aquifer (left) and local natural recharge on the eastern edge of the aquifer by runoff (right); Table S2: Average flow rates and average proportions of STWW in the aquifer from 2017 to 2021 for the baseline model and other models with different parameters for the main flowlines from the different infiltration basins 1, 2, 3 to the Goulot stream (Flowlines 1, 2, 3) and from the Goulot stream to the coastline (Flowline 4).

**Author Contributions:** Conceptualization, Q.G., G.P.-C. and D.I.; methodology, Q.G., G.P.-C.; software, Q.G., C.T., G.P.-C. and D.I.; validation, G.P.-C., D.V., W.K. and J.-M.M.; formal analysis, Q.G., N.D., F.A.M., C.T. and D.I.; investigation, N.D. and F.A.M.; data curation, Q.G., G.P.-C. and D.I.; writing—original draft preparation, Q.G.; writing—review and editing, G.P.-C., D.V., F.A.M., N.D. and D.I.; visualization, Q.G.; supervision, W.K., J.-M.M. and M.P.; project administration, G.P.-C.; funding acquisition, G.P.-C. and M.P. All authors have read and agreed to the published version of the manuscript.

**Funding:** This research was funded by EU Water JPI (Joint Programming Initiative "Water Challenges for a Changing World") AquaNES (Demonstrating synergies in combined natural and engineered processes for water treatment system under grant agreement no. 689450) and EviBAN (ANR-18-WTW7-0008-ERA-NET Cofund WaterWorks—2018).

**Data Availability Statement:** The data that support the findings of this study are available from the corresponding author (Q.G.) upon reasonable request.

**Acknowledgments:** This study has been performed in the frame of the EU Water JPI (Joint Programming Initiative "Water Challenges for a Changing World") within the research projects EviBAN (evidence-based assessment of NWRM for sustainable water management—ANR-18-WTW7-0008-ERA-NET Cofund WaterWorks—2018) and AquaNES (Demonstrating synergies in combined natural and engineered processes for water treatment system under grant agreement no. 689450). We gratefully thank Aubéry Wissocq for initiating the first ideas and designs of the conceptual model and the numerical groundwater flow model, and Marinus Kluijver for performing English corrections of the manuscript, then Eric Dufour, Mickaël Gosselin, Didier Alain (SAUR), and Denis Neyens (Imageau) for their field support and sharing their data, as well as Yan Lefevre (BRGM) and the municipality of Agon-Coutainville, who facilitated the implementation of the monitoring and data sharing.

**Conflicts of Interest:** The authors declare no conflict of interest.

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
