# Peer review of "Multi-Annual Dynamics of a Coastal Groundwater System with Soil-Aquifer Treatment and Its Impact on the Fate of Trace Organic Compounds"

_water, doi:10.3390/w15050934_

Round 1

Reviewer 1 Report

The manuscript title:   multi-annual dynamics of a coastal groundwater system with soil-aquifer treatment and its impact on the fate of Trace Organic Compounds.
It is a good paper. I enjoy for reading this paper. It is acceptable after minor revision.

Comments:

1.      The "Abstract" section: Some important numerical findings deal with the results that should be provided in the Abstract. I think that the presentation of "Abstract" should be reorganized in a systematic and scientific manner.

2.      Please add the "Nomenclature" part to the MS

3.      please use some of the new references such as:

-        Bahrami, M., Zarei, A.R. and Rostami, F., 2020. Temporal and spatial assessment of groundwater contamination with nitrate by nitrate pollution index (NPI) and GIS (case study: Fasarud Plain, southern Iran). Environmental Geochemistry and Health42(10), pp.3119-3130.

-        Mokarram, M., Mokarram, M.J., Zarei, A.R. and Safarinejadian, B., 2017. Using adaptive Neuro-Fuzzy network (ANFIS) to predict underground water quality in the west of Fars province during 2003 to 2013 period. Iranian journal of Ecohydrology4(2), pp.547-559.

Author Response

Response :

  1. The abstract section has been slightly modified in order to find numerical results.
  2. A nomenclature section has been added to supplementary material.
  3. We improved the references citations and the reference section. The first reference (Bahrami et al.) relative to nitrate is not added because our works is not related to nitrates and the reference did not seem relevant to the conceptualization of flows in coastal aquifer or soil aquifer treatment topic. The second one (Mokarra et al.) is not understandable for us because it is not in English but in Arabic.

Reviewer 2 Report

Manuscript is too long and needs to be reduced in length to 2/3 of the current version.

Discussion is a mix between discussion and intro. Needs major work.

Manuscript needs to be check for plaigraism for many sentences.

Author Response

  • Response:

    • A major work has been done to take into account all the detailed review given in the report that was very useful to improve our paper.
    • English revision has been made and allowed to shortened the manuscript. The document structure has been modified as suggested by the reviewer report in order to facilitate the read. In addition, numerous minor corrections has been taken into account in the manuscript.
    • As suggested by the reviewer, discussion and introduction has been clarified. The highlighted part in the discussion has been transferred in the introduction.
    • Considering the plagiarism, the plagiarism detected by the reviewer concern our work from previous paper, the sentences has been synthetised and the sources references has been cited in the text.
    • In addition to the minor corrections, minor corrections suggested by the reviewer has been taken into account. An extensive editing of English language style has been subcontracted. The subcontractor has been added in the acknowledgment section.

Reviewer 3 Report

Comments: Processes-2148603

In the current exploration, the authors examined the Study on the Influencing Factors of the Response Characteristics of the Slide Valve Type Direct Acting Relief Valve with External Orifice. The topic of study is interesting and well-developed. I advise the publication of the manuscript after some minor changes.

·         Please add some very recent papers related to your study.

·         Please mention the novelty of the problem compared to the earlier published work.

·         If possible add the qualitative results in the abstract.

·         What is the advantage of AMESIM simulation compared to others?

·         If possible compare your results with experimental or theoretical works published earlier.

On line 203, in Table 1, the authors used (1) as indicating reference, if so please use a square bracket.

Author Response

The 3rd review does not concern this article (there must be an error) and the 4th one does not specify the requested improvements. A mail has been sent to the editor.

Reviewer 4 Report

The authors undertook the Multi-annual dynamics of a coastal groundwater system with soil-aquifer treatment and its impact on the fate of Trace Organic Compounds. The manuscript quit interesting, the developed hydrodynamic model might be helpful in coastal water treatment. 

Author Response

Responses:

  • An English revision has been made and allowed to shortened the manuscript. The document structure has been modified as suggested by the reviewer report in order to facilitate the read. In addition, numerous minor corrections has been taken into account in the manuscript.
  • In addition to the minor corrections, minor corrections suggested by reviewers has been taken into account. An extensive editing of English language style has been subcontracted. The subcontractor has been added in the acknowledgment section.
  • As suggested by the reviewer, discussion and introduction has been clarified. The highlighted part in the discussion has been transferred in the introduction.
  • In addition to the minor corrections, minor corrections suggested by reviewer shas been taken into account. An extensive editing of English language style has been subcontracted.
  • Some figure has been improved and some legends has been clarified.
  • References section has been improved.

Round 2

Reviewer 2 Report

major changes have been done.